# An NMR sample preparation case study: Considerations for the self-destructive protease caspase-6

Nathanael J. Kuzio[1], Marco Tonelli[2], Jasna Fejzo[3], Jeanne A. Hardy[1,4]*

1 Department of Chemistry, University of Massachusetts, Amherst, Massachusetts, United States of America, 2 National Magnetics Resonance Facility at Madison, Biochemistry Department, University of Wisconsin-Madison, Madison, Wisconsin, United States of America, 3 Biomolecular NMR Core Facility, Institute for Applied Life Sciences (IALS), University of Massachusetts Amherst, Amherst, Massachusetts, United States of America, 4 Models to Medicine Center in the Institute for Applied Life Sciences, University of Massachusetts, Amherst, Massachusetts, United States of America

* jhardy@umass.edu

## Abstract

Proteases represent a difficult family of proteins to purify, concentrate and store at homogeneity due to their toxicity during overexpression and their propensity to self-cleave, leading to the loss of sample stability and function. A protease of interest, caspase-6, is a member of the apoptotic family of caspases, and has been shown to be involved in human neurodegenerative diseases such as Alzheimer's disease and Parkinson's disease. Previous studies have elucidated key structural aspects and potential inhibition mechanisms of caspase-6 through various structural biology techniques such as x-ray crystallography and hydrogen-deuterium exchange mass spectrometry. However, caspase-6 undergoes a structural transition that requires atomic-resolution insight in solution to understand the conformational transitions and ensemble. This can be most optimally achieved using multi-dimensional biomolecular NMR. Prior attempts to study caspase-6 by NMR have failed due to challenges in sample preparation and insufficient protein concentration. Here, we document our exploratory strategy, which ultimately led to the refinement of crucial sample preparation steps and enabled us to obtain isotopically-labeled caspase-6 in yields suitable for heteronuclear NMR studies. We present this work in the hope that it will assist others in the preparation of difficult protein samples, particularly proteases.

## Introduction

Caspases comprise a family of human proteases involved in both healthy cell homeostasis as well as a number of disease states. Members of the caspase family cleave immediately after aspartate residues in their substrate proteins, using a catalytic cysteine-histidine dyad [1–3]. Originally referred to as mammalian Ced-3 homologue

**Data availability statement:** All relevant data are within the paper and its Supporting information files.

**Funding:** NK was funded by National Institutes of Health T32 GM135096. The Hardy lab (JH) was supported during various phases of this project by NIH R01 GM080532 and NIH R35 GM149348. The funders had no role in study design, data collection and analysis, decision to publish, or preparation of the manuscript.

**Competing interests:** The authors have declared that no competing interests exist.

2 (Mch2), caspase-6 (casp-6) was discovered as a human homologue to the cell death machinery found in *Caenorhabditis elegans* [4]. Around the same time, caspases began being categorized into subgroups based on sequence similarity [5], substrate specificity [2,3], and their involvement in the apoptotic cascade [6,7]. Casp-6 along with caspases-3 and 7 are classified as effector or executioner caspases due to their activation by initiator caspases and downstream role in carrying out programmed cell death [8].

Previous studies [9–12] have shown casp-6 involvement in neurodegeneration in the context of Alzheimer's Disease. Tau cleaved at known casp-6 cleavage sites as well as active casp-6 are present in the brains of deceased Alzheimer's Disease and frontotemporal dementia patients [13,14]. Additionally, colocalization of active casp-6 with phospho-tau suggests a possible correlation between these two Alzheimer's Disease hallmarks [13]. Data indicating a potential causal role in neurodegeneration has led the field to identify casp-6 as an attractive drug target for these diseases. Thus, we have undertaken research investigating the structure and potential inhibition mechanisms of the protease.

We have previously used techniques such as x-ray crystallography and hydrogen-deuterium exchange mass spectrometry (HDX-MS) to identify key structural regions of casp-6 that set it apart from other caspases. These unique structural characteristics, such as the helix-strand interconversion in the 130's region [15–17], may render it susceptible to selective inhibition over structurally similar family members. In addition, physiological regulators such as phosphorylation [18–20], metals [21], and nucleotides [22], exosite patches [23], tumor-associated mutations [24], and mutations found in healthy humans also impact casp-6 function positively and negatively [25]. Casp-6's involvement in neurodegenerative diseases has positioned it as a target for synthetic inhibitors [26–32] as well. In all these cases, the ability to assess inhibitor effects on this unique interconversion would be invaluable for further drug development. While x-ray crystallography provides atomic-level detail of some discrete conformations and HDX-MS provides peptide-specific dynamics, neither of these techniques can provide atomic-level detail on the dynamics of the protein in solution. For these reasons, we aim to utilize nuclear magnetic resonance (NMR) spectroscopy to assess casp-6 structure and dynamics, as this is the only technique capable of measuring dynamics occurring on a residue-specific level.

NMR spectroscopy is uniquely valuable for its ability to provide residue-specific information dependent upon its chemical environment. Perturbing and observing the relaxation of nuclear spins in a bulk magnetic field returns a unique peak signature for a given conformation [33], making NMR an exceptional technique for the study of dynamics [34–39]. Observing effects on an atom's nuclear spin typically requires incorporation of isotopes with a ½ nuclear spin, for example $^1H$, $^{13}C$, and $^{15}N$ at desired locations. To achieve high signal, NMR samples must be highly concentrated and stable for the duration of the NMR experiment. To prevent signal overlap and broad-spectrum distortion, often samples must be analyzed in a deuterated buffer. This presents a challenge as casp-6 is expressed at low yields [40,41] and is prone to aggregation and precipitation under certain conditions during concentration and

buffer exchange. Like all proteases, at high concentrations casp-6 shows self-destructive tendencies such that samples of active proteases degrade over time. NMR studies have been possible on caspases, including casp-8 [42] and −9 [43], which are monomeric and inactive prior to activation on an activating platform or in their inactive zymogen state [42]. Casp-6 is constitutively dimeric and key studies of casp-6 require study of the catalytically active enzyme, so approaches that have worked previously on other caspases did not work on casp-6. In addition, the solubility properties of dimeric casp-6 and its ability to be expressed differ from casp-8 and −9. While prior work has addressed various approaches for preparation of NMR-active samples generally, these studies were not sufficiently predictive to render our preparation of casp-6 samples routine or seamless [44–47]. In particular, common concentration and buffer exchange methods using spin filters have been unreliable as this often leads to percent recoveries of <10% for casp-6, especially when working with concentrations below 40 µM. For these reasons we found it necessary to undertake a thorough exploration of approaches to produce samples suitable for robust analysis by NMR. Here, we record our exploratory approach which ultimately allowed optimization of key steps in the sample preparation process (Fig 1) to obtain isotopically-labeled casp-6 in modest, yet sufficient, yield for a variety of [$^1$H-$^{13}$C] NMR experiments.

## Results

### Casp-6 can be satisfactorily overexpressed in *E. coli* in 2x M9 minimal media

A first and necessary step toward performing NMR analysis on casp-6 is preparation of sufficient quantities of isotopically-labeled protein. For preparation of non-labeled samples, casp-6 is typically overexpressed in an *Escherichia coli* expression system in nutrient-rich (2xYT) media (Fig 2A, top). Large 2xYT cultures are inoculated with a 1,000-fold dilution of a dense seed culture and grown at 37°C to an optical density at 600 nm ($OD_{600}$) of 0.6–0.9 before protein production is induced with 1 mM isopropyl β-D-1-thiogalactopyranoside (IPTG) for 18 hours at 20°C [17]. This generates yields of 1 milligram pure protein/liter culture (Fig 2B). For isotopic labeling, minimal media provides a single, controllable carbon source and single controllable nitrogen source, ensuring the desired isotopes are incorporated during protein expression. Typical carbon sources are glucose or pyruvate and nitrogen sources are ammonium chloride or ammonium sulfate [48,49] (S1 Table). In contrast to the growth process in rich media, the large cultures of minimal media are inoculated with a much larger dilution (4,000-fold) of a starter culture grown in LB media to limit contamination of the minimal media with components of the rich media. Cultures in minimal media were grown to an optimized $OD_{600}$ at 30°C and subsequently induced with IPTG at 20°C (Fig 2A, bottom). In addition to the $OD_{600}$, we tested the impact of two media conditions referred to as M9 and 2x M9. Previous studies [50,51] have reported that increasing the concentration of the phosphate salts and thus increasing the buffering capacity of the media allows a higher $OD_{600}$ to be reached without as great of a toxic effect from acidic byproducts from *E. coli* growth. This has led to greater protein yields in 2x M9 compared to M9 minimal media as the *E. coli* remains more viable at a higher $OD_{600}$.

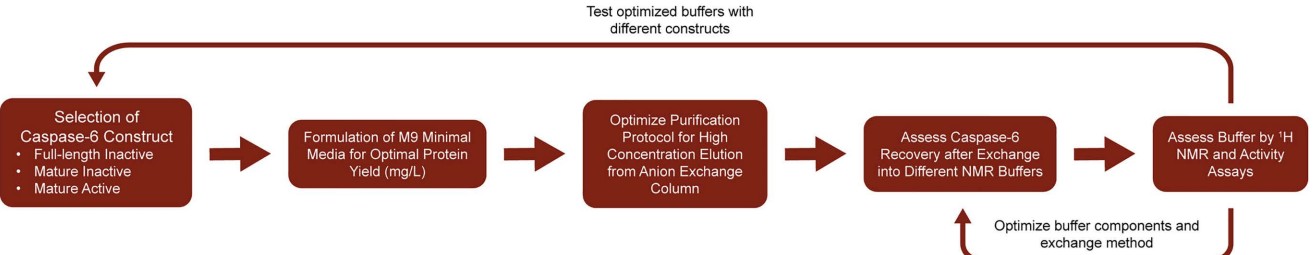

**Fig 1. Workflow of casp-6 sample preparation and optimization for NMR spectroscopy.** Each step in the protein production, purification, and exchange process was explored and assessed by multiple tests. Key areas of manipulation are noted.

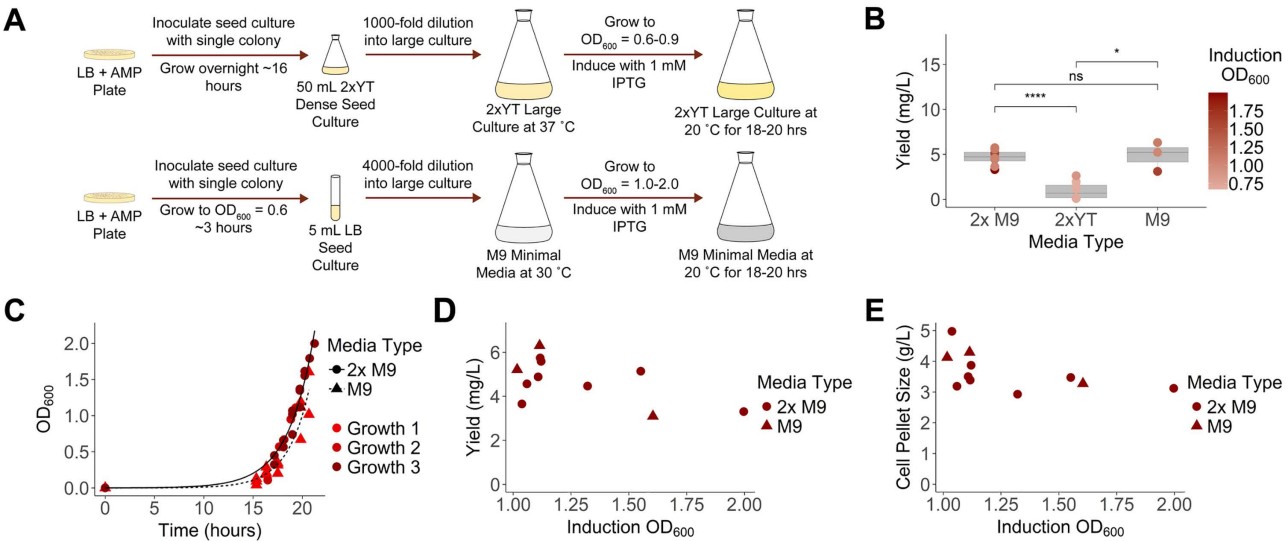

**Fig 2. Determination of ideal *E. coli* growth conditions for casp-6 overexpression in minimal media. (A)** Bacterial cell culture process comparing growth of BL21 (DE3) *E. coli* in M9 (M9) or 2x M9 minimal media (2x M9) to 2xYT rich media. **(B)** Protein yield was dependent on culture media used (2x M9: n = 8, 2xYT: n = 8, M9: n = 3 independent purifications). Casp-6 yields were calculated following a standardized purification protocol. **(C)** Growth rates of BL21 (DE3) *E. coli* were similar in M9 (triangles) and 2x M9 (circles) minimal media in three independent experiments. **(D)** Protein yield following purification was measured as a function of induction OD$_{600}$. **(E)** Bacterial pellet mass was measured as a function of induction OD$_{600}$.

We did not observe significant differences in the growth time required to reach a given OD$_{600}$ following the standard 4,000-fold dilution in traditional M9 vs 2x M9 minimal media (Fig 2C). We induced protein expression with 1 mM IPTG at OD$_{600}$ values ranging from 1.0–2.0. Interestingly, protein yield was approximately 5 mg/L on average for cultures grown in M9 or 2x M9 minimal media, which was significantly higher than 1 mg/L which was observed for those grown in 2xYT, a rich media (Fig 2B). This was unexpected as expression levels often decrease in less nutrient-rich minimal media. We hypothesize that casp-6 can be expressed in minimal media because a higher OD$_{600}$ can be induced without the production of by-products of *E. coli* growth often observed in 2xYT. Another possible explanation is that the nutrient-limited M9 media yields less leaky expression at the *lac* operon, thus limiting buildup of casp-6, which, due to its proteolytic activity, potentially inhibits robust growth in *E. coli*. For both M9 and 2x M9 media an induction OD$_{600}$ of 1.1 yielded the greatest amount of protein produced per liter at 5–6 mg/L, with yield dropping to 3–4 mg/L as induction OD$_{600}$ increased (Fig 2D). Increasing induction OD$_{600}$ also corresponded with an unexpected decrease in cell pellet size from 4 g/L to 3 g/L, suggesting the decreased protein yield may also be a result of cell division decreasing and cells entering stationary phase prior to induction at higher cell densities. It is possible that some cells may also die or become senescent as the culture becomes overcrowded (Fig 2E). We noted that contaminants following purification did not appear to change as a function of induction OD$_{600}$ (S1 Fig). Despite no significant difference in the overall amount of protein that was expressed in M9 or 2x M9 minimal media with both compositions yielding approximately 5 mg/L (Fig 2D), 2x M9 minimal media appeared more tolerant to induction at a higher OD$_{600}$, likely due to the increased buffering capacity. Based on these data, the optimal conditions for casp-6 overexpression consisted of growth in 2x M9 minimal media to an OD$_{600}$ of 1.1 followed by induction for 18 hours at 20°C.

## Elution concentration improved by optimizing flow rates and gradients during purification

A key requirement is to obtain highly concentrated casp-6 for buffer exchange into appropriate conditions for subsequent experiments. Our lab has observed that at casp-6 concentrations below 40 µM spin concentration of the protein leads

to significant sample loss possibly due to coating of the concentrator apparatus enabled by low solution viscosity. Thus, our goal was to obtain a suitably high casp-6 concentration for subsequent manipulation directly from the purification procedure, without sacrificing purity. Historically, we and others have purified casp-6 via nickel-affinity (Ni-affinity) chromatography followed by anion-exchange chromatography [15–17,19,21–25,28,29,32,52] or size exclusion chromatography [20,53–55]. We modified the anion-exchange purification protocol to optimize the casp-6 elution profile and thereby increase the concentration of casp-6 obtained. Flow rate and fraction volume were also optimized to increase concentration and allow for selection of the most concentrated fractions, respectively.

Removal of impurities and increased concentration through rapid elution of casp-6 in a smaller volume was achieved through elution of casp-6 from the ion-exchange resin in a single-step increase in NaCl concentration instead of a linear gradient (Fig 3A, 3B). The optimal elution profile was found to be a linear gradient to 150 mM NaCl followed by a step increase to 230 mM NaCl, as in Protocol 4 (Fig 3B). Reducing the flow rate from 5 to 1 mL/min increased the protein concentration in the eluted fractions (Fig 3B), likely due to longer resin-buffer contact time.. Combining the improved step elution profile (230 mM NaCl), this lower flow rate (1 mL/min) and smaller fraction volumes (0.5 mL) permitted selection of more concentrated samples (Fig 3B) of ~100 µM instead of 40 µM which was achieved with the traditional elution protocol. This increase in elution concentration was essential for successful buffer exchange without significant sample loss.

### pH impacts casp-6 purification and yield

For NMR spectroscopy, low pH is optimal for many experiments such as $^1$H-$^{15}$N HSQC because it decreases the rate of amino-proton exchange with bulk water. In typical casp-6 purification protocols, pH 8.5 buffers have been used for both Ni-affinity and anion-exchange due to the casp-6 pI of 8.0. We asked whether yields of casp-6 could be further increased by performing purification at low pH or by dilution into a low pH buffer after Ni-affinity purification at pH 8.5. When Ni-affinity and cation-exchange chromatography were performed at pH 7.5, the yields were <10% of those observed when following the traditional anion-exchange protocol. We next performed Ni-affinity purification in Tris pH 8.5 buffer and then diluted the casp-6 obtained with HEPES pH 7.5 and attempted cation-exchange. Robust binding to the cation-exchange resin was not observed, leading to poor yields of casp-6 at concentrations of <10 µM. This contrasted with a report from another group that addition of acetic acid could be tolerated [53]. Nevertheless, we observed better yields and purification using pH 8.5 buffers and concluded that robust purification required this higher pH.

### Omitting anion-exchange purification increased casp-6 concentration without impacting purity

Following the decreased yield observed with cation-exchange chromatography, we sought to address the necessity of an ion-exchange purification step. Although we successfully optimized the anion-exchange purification conditions, and this purity may be useful for other experimental endpoints, we ultimately concluded that the purity obtained by solely a nickel-affinity purification and buffer exchange yielded samples of sufficient purity for NMR spectroscopy. This was supported by the adequate purity of the nickel elution fractions ("Ni Elu", Fig 3C) and the inability of anion exchange to remove higher molecular weight contaminant proteins ("QFT" and "Q Wash", Fig 3C). Additionally, using solely a Ni-affinity purification (Fig 3D) followed by buffer exchange with a PD-10 desalting column (Fig 3E) resulted in an increased growth and purification yield of ~1 mg/L compared to ~0.5 mg/L that were obtained from a Ni-affinity and anion-exchange purification. Thus, subsequent purification protocols omitted the anion-exchange purification step entirely to retain as much casp-6 as possible. This simplified protocol (S1 Protocol) also produced a higher starting concentration of casp-6 for the buffer exchange step, which routinely failed at low casp-6 concentrations.

### Buffer exchange by desalting column provides the most efficient recovery

For NMR experiments it is essential that the proteins be analyzed in deuterated buffers to allow observation of protein proton signal without overlap with proton signal from the buffer components. Thus, in most NMR-directed protein production

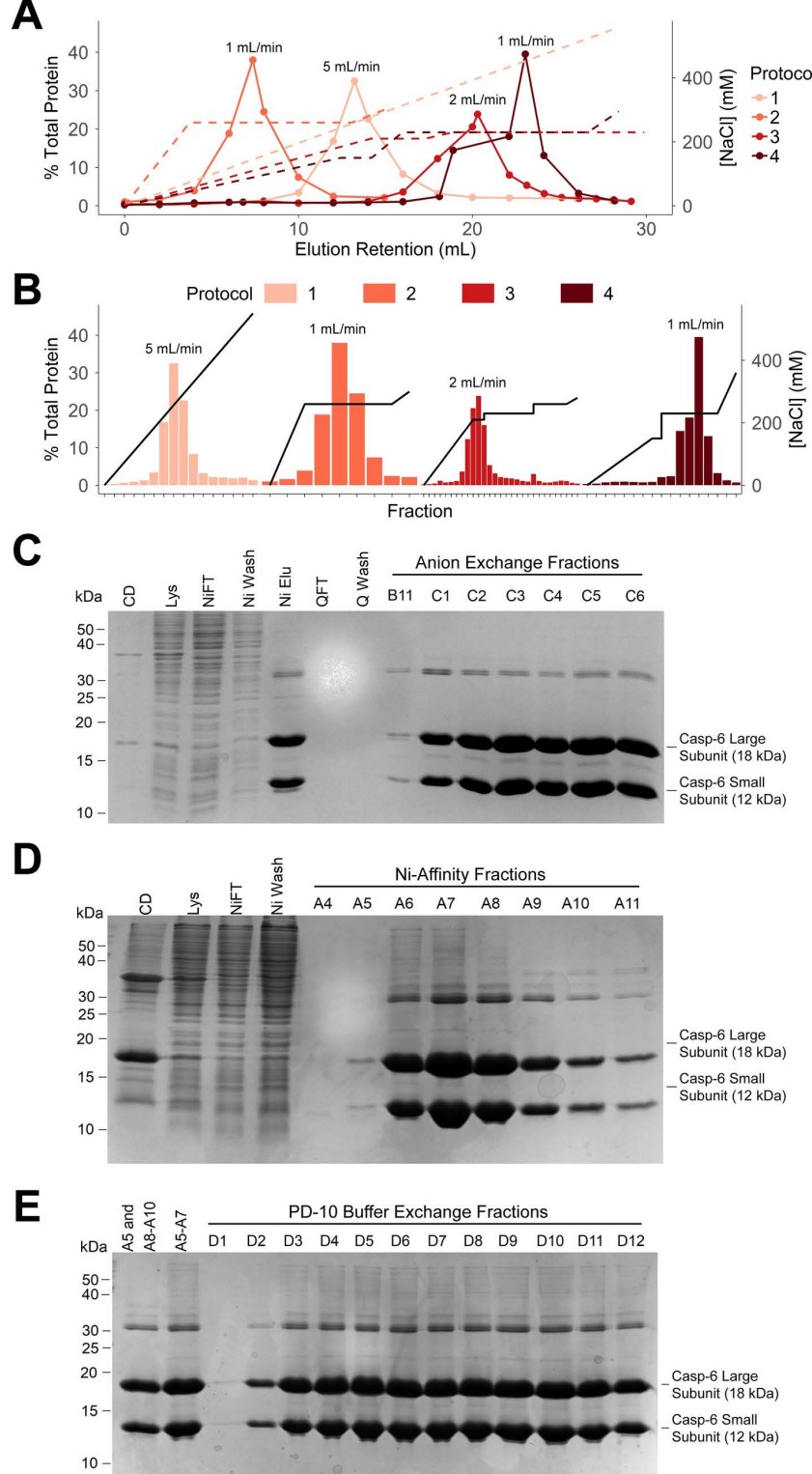

**Fig 3. Comparison of anion-exchange purification protocols to balance elution concentration and purity. (A)** Overlay of four anion-exchange chromatograms on a 5mL HiTrap Q column (Cytiva Life Sciences) tracked by NaCl concentration dashed lines, left Y axis represents percent of the total protein obtained for each purification scheme tested. **(B)** Comparison of four anion-exchange protocols tested illustrates benefits of lowered flow rate

and step elution profile in both % Total Protein and NaCl concentration in the elution profile (black line). Fraction numbers excluded for clarity. **(C)** Purity of casp-6 following anion-exchange was assessed by SDS-PAGE and did not significantly improve from that obtained from Ni-affinity chromatography on a 5 mL HisTrap HP column (Cytiva Life Sciences) when followed by anion-exchange on a 5 mL HiTrap HP Q Anion-Exchange column (Cytiva Life Sciences). Fractions from the purification include cell debris following cell lysis and centrifugation (CD), cell lysate (Lys), Ni column flow-through (NiFT), Ni column wash (Ni Wash), Ni column elution (Ni Elu), anion-exchange flow-through (QFT), anion-exchange wash (Q Wash), as well as individual fractions from anion-exchange. **(D)** Purity of casp-6 following solely Ni-affinity purification was assessed by SDS-PAGE and found sufficient for NMR following buffer exchange. Fractions from the purification include cell debris following cell lysis and centrifugation (CD), cell lysate (Lys), Ni column flow-through (NiFT), Ni column wash (Ni Wash), as well as individual fractions from the Ni-affinity elution. **(E)** PD-10-desalting-column buffer exchange using a high initial concentration of casp-6 yielded a higher final concentration and improved recovery. Two different samples were analyzed. One pool included half of fraction A5 and A8-A10 from the Ni-affinity purification shown in **(D)**. The second pool included half of A5 and A6-A7 from the Ni-affinity purification shown in **(D)**. Individual fractions from the PD-10 desalting column elution are also shown.

protocols, buffer exchange will be a required undertaking. The primary methods of buffer exchange for NMR samples consist of repetitive dilution and concentration steps [56] using a centrifugal filter or via a desalting column [44,57], such as a Cytiva NAP-5 or PD-10 column. Other approaches such as dialysis were deemed unfeasible due to the large volume of deuterated buffer required. Buffer exchange by centrifugal filter was unsuitable due to observed final recoveries <10%. This loss of sample was likely due to the dilution steps lowering the concentration of casp-6 below the 40 µM threshold observed above. Attempts to remedy this included: altering number of dilution and spin repetitions or dilution factor, blocking the filter membrane with 1% BSA in PBS or 5% Tween, decreasing centrifugal force, and frequently mixing concentrate to prevent concentration gradients within the filter, however none significantly improved recovery, thus this method was abandoned.

While buffer exchange via a desalting column also resulted in loss of protein, much less casp-6 was lost than with centrifugal filtration. The desalting column buffer exchange efficiency was dependent on several factors including the size of desalting column being used (Cytiva NAP-5 or Cytiva PD-10), the sample volume to be exchanged, as well as the differences between the starting buffer and exchange buffer components. As expected, varying the sample volume to be exchanged and the size of desalting column impacted the dilution factor as well as percent recovery of exchanged protein. For example, exchanging a volume of casp-6 smaller than the upper limit for the desalting column resulted in a greater percent recovery of exchanged protein but a larger dilution factor. In contrast, exchanging the maximum volume of casp-6 recommended for the column resulted in a lower dilution factor but not all applied protein was eluted in the recommended elution volume. Decreasing pH (starting buffer Tris pH 8.5) and removal of glycerol or DTT resulted in significant loss of ~30% of casp-6 to precipitation (Fig 4A, 4B). Loss of protein was likely due to precipitation near its isoelectric point, resulting in instability or increased non-specific binding to Sephadex resin. The presence of glycerol likely stabilizes casp-6 through crowding and DTT stabilizes the samples by maintaining the reduced state of the six surface cysteines. Ultimately, limiting changes in pH, glycerol and DTT concentrations, as well as choosing a suitable desalting column for the given sample volume were essential for efficient buffer exchange of casp-6. This allowed for efficient buffer exchange with up to 70% recovery, a dramatic improvement from <10% which is seen with traditional centrifugal spin filter methods. These studies employed a Cytiva NAP-5 or PD-10 desalting column (dependent upon sample volume) and exchanged into a buffer at pH 8.5 with 5% glycerol and 10 mM DTT.

## Tris buffer is optimal to balance casp-6 activity, NMR spectral quality, and recovery during buffer exchange

The goal of optimizing an NMR buffer is to maximize the stability of the protein of interest to limit depreciation of NMR signal intensity. Common reagents used in NMR experiments include pH buffering agents, salt, metal chelators, detergents, and reducing agents [45–47]. Buffer components known to negatively impact NMR signal intensity include increasing pH when observing $^1$H-$^{15}$N correlations (via increased exchange of amino-protons with bulk water), increasing ionic strength of the solution (adding noise to the signal collection), and protonated reagents that cannot be deuterated. Enzymatic activity assays as well as 1-dimensional $^1$H NMR spectra were used to evaluate buffering agents and buffer components

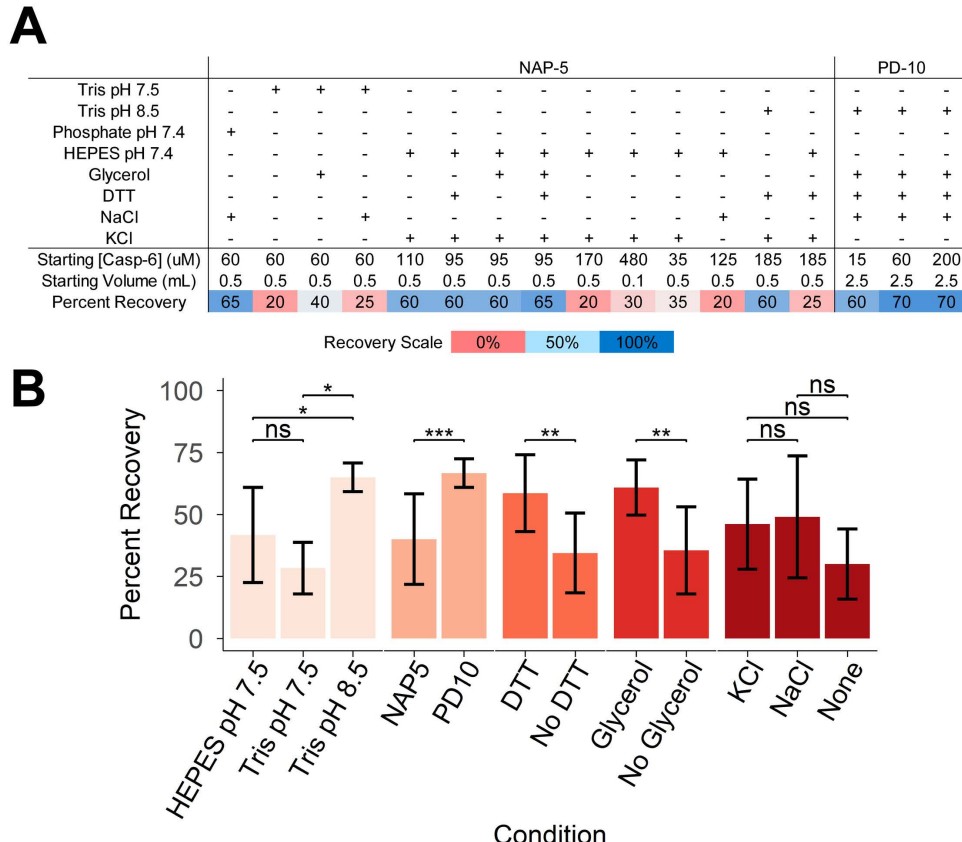

**Fig 4. Effects of buffer components and desalting column size on percent recovery following buffer exchange. (A)** Casp-6 recovery following buffer exchange on NAP-5 or PD-10 Sephadex-G25 desalting columns (Cytiva Life Sciences) under conditions listed was estimated by spectroscopic absorption at 280 nm. **(B)** Cumulative differences between various exchange conditions summarizing the results of the 17 individual exchange protocols tested in (A) assessed buffer and pH, presence of DTT, glycerol or salts and desalting column sizes (*p<0.05, **p<0.01, ***p<0.001, ****p<0.0001). Bars represent mean±standard deviation (n=3-9). Bars are colored by condition category. Together these suggested superior casp-6 recovery in Tris pH 8.5, with DTT and glycerol, using a PD-10 column.

best suited for casp-6 NMR. An increase in enzyme initial velocity ($V_0$) of cleavage of the fluorogenic peptide substrate VEID-amc indicates a properly folded caspase and sufficient protein stability. Dispersion of amide (downfield) and methyl protons (upfield) in 1-dimensional $^1$H NMR spectra provide insight into the folded state of a protein. During this optimization, prior reports on the impact of salt and pH on casp-6 function [1,16] were also considered.

Three buffering agents--HEPES pH 7.5, phosphate pH 7.4, and Tris pH 7.5--and additives were tested in a proteolytic assay using VEID-amc substrate (S2 Table). Casp-6 showed significantly higher activity in HEPES than in either Tris or phosphate buffers and exhibited significantly higher activity in phosphate buffer than Tris buffer (Fig 5A, S3 Table). Interestingly, when combining initial rates across the three buffer types, there were no statistically significant differences between any of the individual buffer additives (Fig 5B). Within buffer groups, beta-octyl-glucoside (BOG) did stand out as increasing the initial rate of casp-6. It is possible that this detergent may provide solubilization of casp-6 and prevent aggregation. Lastly, the effect of varying NaCl concentration as well as pH on casp-6 activity was tested in sodium acetate pH 4.6, phosphate pH 7.4, and Tris pH 7.5 buffers. Overall activities in phosphate and Tris buffers were comparable and significantly higher than that in sodium acetate (Fig 5C). While NaCl concentration did not appear to have an impact in the phosphate buffer samples (possibly due to the already high ionic strength of the sodium phosphate), 120 mM NaCl led

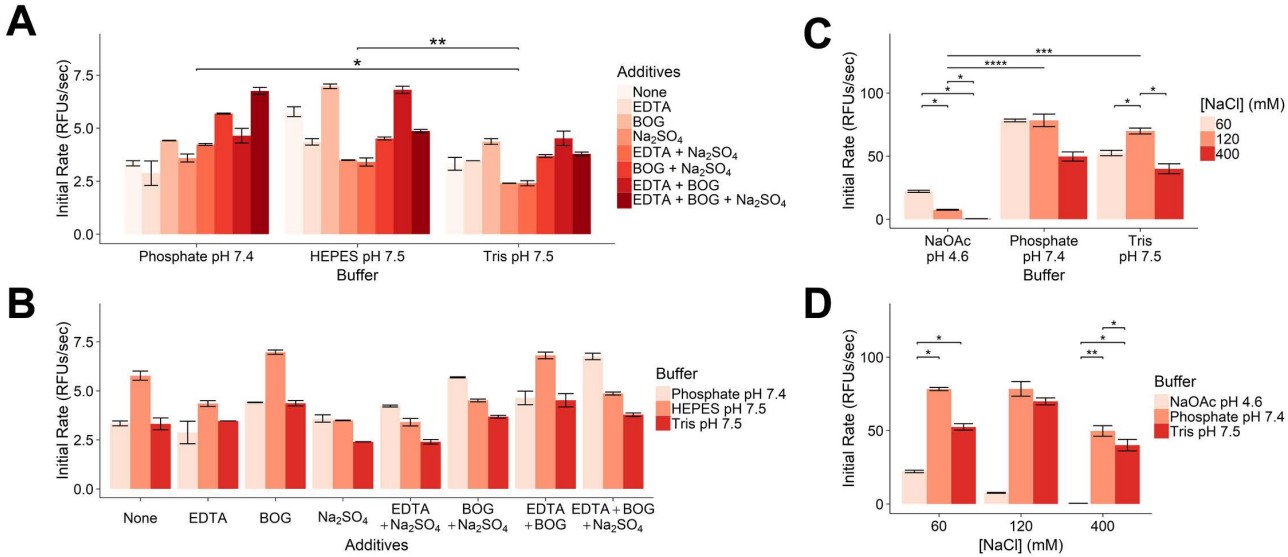

**Fig 5. Impact of buffering agents and additive on casp-6 activity. (A)** Cleavage of the fluorogenic casp-6 substrate VEID-amc was monitored in the presence of various buffer additives and buffering agents. Activity was greater in HEPES pH 7.5 and phosphate pH 7.4 than in Tris pH 7.5. **(B)** Buffer additives and buffering agents showed no statistically significant differences between different buffer additives. **(C)** Statistically significant differences in casp-6 activity were observed between buffering agents and NaCl concentrations within sodium acetate pH 4.6 and Tris pH 7.5 buffer conditions. **(D)** Statistically significant differences were not observed between NaCl concentrations when combining results from various buffering agents. Bars represent mean ± standard deviation. All conditions were tested in two technical replicates and statistical significance is indicated by *p < 0.05, **p < 0.01, ***p < 0.001, ****p < 0.0001.

to significantly higher casp-6 activity than either 60 mM or 400 mM NaCl in in Tris buffer (Fig 5C). Overall, the concentration of NaCl did not appear to significantly change the initial rate of casp-6 activity (Fig 5D), however, these data may be skewed due to the different contributions of each buffer to the overall ionic strength of the sample as well as the changes in pH.

The impact of several of these buffer components such as buffering agent, NaCl concentration and BOG were tested by $^1$H NMR as well. Focusing on the dispersion of amide proton and methyl proton peaks, Tris and HEPES buffers provided a better resolved spectrum and higher signal to noise than phosphate buffer. The differences in spectral quality between Tris and HEPES buffers were minimal (Fig 6A). Taken together with the improved recovery observed in Tris buffer, it was concluded that Tris pH 8.5 would be the preferred buffer for these NMR experiments despite casp-6 exhibiting lower VEIDase activity under these conditions. The ideal NaCl concentration should balance sufficient salt for casp-6 to remain folded without decreasing the signal intensity in the $^1$H NMR experiment. Concentrations up to 240 mM NaCl maintained the maximal peak intensity, suggesting that even with multidimensional NMR experiments concentrations up to 200 mM should be tolerated (Fig 6B). Additionally, the lack of signal deterioration with increased NaCl concentration in the $^1$H NMR spectra and trends observed for the Tris VEIDase assays (Fig 5C) is consistent with a previous study that had shown similar concentrations of NaCl were required to stabilize casp-6 [16]. Given that BOG was the only additive that appeared to increase casp-6 activity, unlabeled BOG was added to an NMR sample to assess spectrum enhancement. While the presence of BOG slightly improved signal in the amide region (Fig 6C), we concluded that the cost associated with exchanging the protein into buffer containing deuterated BOG and the lack of VEIDase improvement (Fig 5B) outweighed the benefit seen in the $^1$H NMR spectrum. Glycerol also did not appreciably improve spectral quality (Fig 6D) for fully mature constitutively two-chain casp-6 [16] (Casp-6 D179 CT) nor the catalytically inactive full-length casp-6 (Casp-6 FL C163S). The amide region of casp-6 FL C163S provided sharper peaks and more dispersion than casp-6

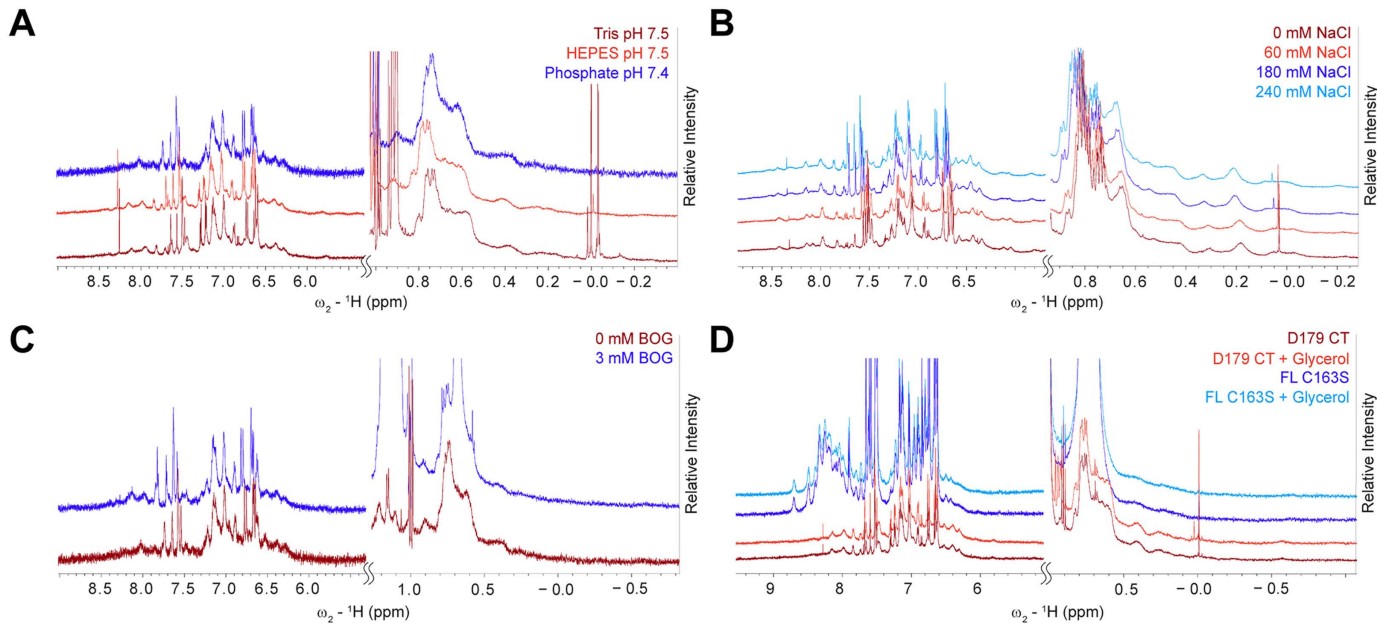

**Fig 6. ¹H NMR spectral quality as a function of varying buffer conditions. (A)** Impact of buffering agents Tris pH 7.5, phosphate pH 7.4, and HEPES pH 7.5 was assessed by monitoring amide and methyl peak dispersion in the casp-6 D179 CT ¹H NMR spectrum measured at 600 MHz. **(B)** Increasing NaCl concentration resulted in <5% variation in peak height or linewidth in the casp-6 D179 CT ¹H NMR spectrum. **(C)** Presence of octyl-β-glucoside (BOG) did not significantly improve casp-6 D179 CT ¹H NMR peak resolution. **(D)** Casp-6 FL C163S showed superior dispersion of amide peaks while the spectrum of casp-6 D179 CT showed greater methyl peak dispersion. The presence of glycerol did not produce notable spectral improvement for either casp-6 construct.

D179 CT. This may be due to the presence of a flexible linker and prodomain in the full-length construct, which is cleaved and removed upon zymogen maturation and is also absent in the CT version. It is expected that these prodomain and linker residues would give sharper peaks in casp-6 FL C163S due to their increased mobility. The mature form of casp-6 provided superior peak signal in the methyl regions of the spectra suggesting this construct may be more suitable for methyl-focused NMR experiments. Subsequent stability tests on methyl-labeled casp-6 further supported the optimization showing no significant reduction in peak intensity for up to five days when incubated at 25°C (S2 Fig). Harnessing the sensitivity of methyl-specific labeling as well as the optimized sample preparation methods outlined above, we are now able to use ¹H-¹³C heteronuclear multiple quantum coherence (HMQC) spectra of Ileδ1-¹³CH₃ casp-6 D179 CT to investigate the unique dynamics and regulation of this neurodegenerative-disease-associated protease.

## Summary and conclusions

The goal of these studies has been to determine the most efficient method for casp-6 NMR sample production. Steps in this process were individually optimized to increase yield of labeled casp-6, ensure sufficient purity, and improve resolution and signal intensity in the resulting NMR spectra. Some changes to the sample preparation protocol improved one aspect but simultaneously worsened other aspects. In the end, we found it necessary to exclude otherwise productive steps to balance the overall goals to maximize protein quality as enumerated in our protocol (S1 Protocol).

Ultimately, the method of sample preparation chosen for casp-6 consists of growth and isotopic-labeling in 2x M9 minimal media, followed by Ni-affinity purification, then buffer exchange via a desalting column into a deuterated NMR buffer that resembles the previously used anion-exchange buffers (Fig 7). We chose to use 2x M9 minimal media rather than regular M9 minimal media because the higher buffering capacity increased the tolerance to induction at higher levels of culture density ($OD_{600}$ values). While the optimized $OD_{600}$ did not show significantly different yields between the two media

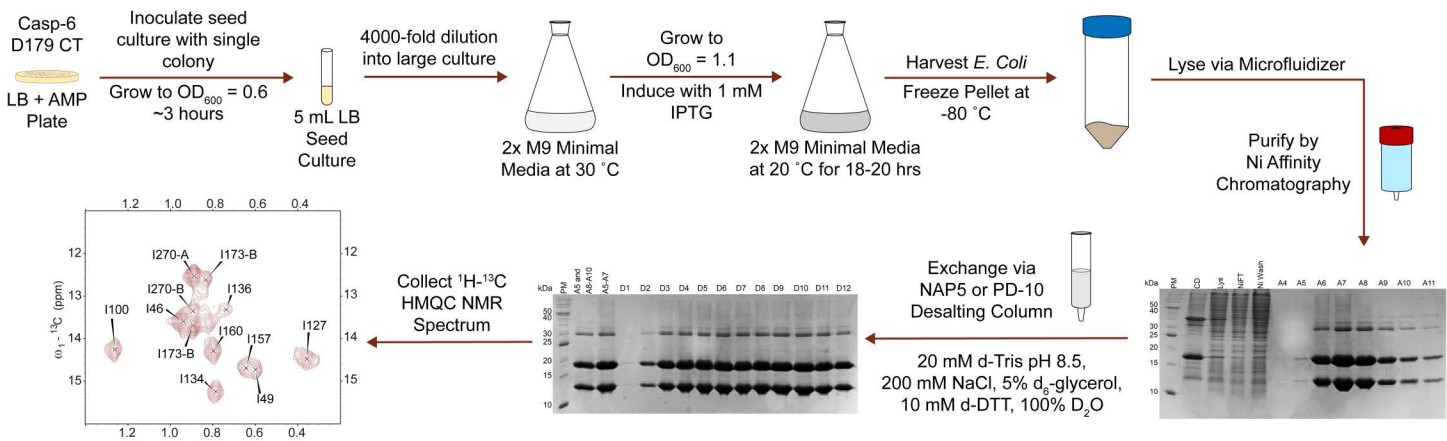

**Fig 7. Optimized NMR sample preparation protocol for casp-6 D179 CT.** The optimal preparation protocol for casp-6 NMR consisted of growing a large culture of casp-6 D179 CT in 2x M9 to an $OD_{600}$ of 1.1 prior to induction, purification by Ni-affinity chromatography, and buffer exchange via a PD-10 desalting column into a buffer containing 20 mM d-Tris pH 8.5, 200 mM NaCl, 5% d-glycerol, 10 mM d-DTT, and 100% $D_2O$. NMR assignments for the $^{13}$C-Ileδ1-methyl labeled casp-6 D179 CT $^1$H-$^{13}$C HMQC spectrum can now be found at BMRB entry 53341 in the biological magnetic resonance bank.

compositions, increased tolerance to high culture density is valuable as variable growth rates have been observed. As noted previously, the anion-exchange purification step was eliminated due to the small increase in purity and decreased recovery that were obtained by this step. We concluded that a Ni-affinity purification followed by buffer exchange provided sufficient purification of the sample with greater protein recovery. Additionally, since the Ni-affinity purification eluted the protein with a step elution, much higher concentrations of casp-6 were obtained directly from the purification compared to those eluted from an anion-exchange column even with the optimized elution protocol. This meant that less concentration via a centrifugal spin concentrator was required prior to buffer exchange into the NMR buffer conditions.

It should also be noted that choice of buffer exchange via a NAP-5 (0.5 mL max load volume) or a PD-10 (2.5 mL max load volume) desalting column was dependent upon the amount of protein being purified and consideration of the following factors: 1) a large reduction in volume via spin concentrator often led to aggregated or precipitated protein and loss of sample even when starting with concentrations above the recommended 40 µM, 2) a higher concentration of casp-6 loaded onto a desalting column resulted in more concentrated sample being eluted, and 3) too high of a concentration loaded onto the desalting column (especially pertinent for NAP-5) led to a significant portion of the protein eluting after the recommended elution volume and thus not being fully exchanged. Given the proteolytic nature of casp-6 we recognized the need for rapid purification and storage. Ultimately, we observed the greatest loss of casp-6 was associated with spin filtration.

Our overall method limited spin filtration and minimized the time between cell lysis and freezing of purified NMR samples of casp-6. The most effective protocol we found for casp-6 production and exchange started with growing large volumes of culture to minimize the need for concentration prior to the desalting step. The size of culture used in preparation was determined by the final use case. We optimized two protocol batch sizes that each produced pure, stable protein. For experiments requiring a final concentration not greater than 130 µM casp-6, a 4 L culture was optimal. Culturing 4 L in 2x M9 allowed Ni-affinity column purification of 4–4.5 mL of 100−130 µM casp-6. Casp-6 at this concentration could be further concentrated to 2.5 mL of ~200 µM casp-6 by a single spin filtration without significant loss of material. After PD-10 desalting, 2–2.5 mL of 100−130 µM casp-6 was obtained. Alternatively, if concentrations of 200 µM casp-6 or greater were required, an 8 L culture in 2x M9 media was optimal. 8 L of culture produced approximately 5 mL of casp-6 at 200

µM casp-6 from the Ni-affinity column. Such samples could be further concentrated for a higher concentration exchange yielding samples at approximately 200 µM casp-6 post exchange. However, note that concentrating to above 200 µM does sometimes result in significant precipitation. We have not yet determined the physical basis for this phenomenon. We found that volumes above 5 mL of 200 µM casp-6 samples could be split and exchanged on two separate PD-10 columns without significant sample loss. This approach was useful in the case of pre-exchange incubation with inhibitors. We observed that for protein expressed from smaller culture volumes, a NAP-5 exchange could be used, however, final observed concentrations rarely exceed 80 µM and recoveries tended to be lower. Lastly, exchanged fractions at too low of a concentration or a sample that was eluted from a Ni-NTA column that has not been exchanged can subsequently be pooled with similar samples from other purifications and repurified or concentrated and exchanged later.

The last step in the sample preparation process was to test the quality of NMR spectra and reassess the protocol with different casp-6 constructs. We considered two cleavage states of casp-6: full-length inactive casp-6 FL C163S, wherein the catalytic cysteine was replaced by serine, and the fully mature constitutive-two chain construct casp-6 D179 CT that produces protein that mimics the fully mature protein due to separate expression of the large and small subunits. Upon overexpression in *E. coli*, the prodomain is removed by self-cleavage. While the initial ${}^{1}$H NMR spectra suggested casp-6 FL C163S might be superior for observing ${}^{1}$H-${}^{15}$N correlations of the amide backbone, these sharp peaks were likely in the disordered prodomain and flexible intersubunit linker of casp-6. Observing ${}^{1}$H-${}^{13}$C correlations of methyl groups using the casp-6 D179 CT construct emerged as a better option for investigating residues in the core of the protein. Given that we are particularly interested in a structural interconversion occurring in the protein core, we decided to focus our attention on the ${}^{1}$H-${}^{13}$C HMQC spectrum of the 12 Ileδ1-${}^{13}$CH$_3$ groups of casp-6 D179 CT (Fig 7). The ${}^{1}$H-${}^{13}$C HMQC spectrum of Ileδ1-${}^{13}$CH$_3$-labeled casp-6 showed 13 isoleucine peaks, which was expected as some Ile residues are known to undergo conformational exchange. Additionally, modest signal-to-noise, well-dispersed chemical shifts, and linewidth were consistent with a stable and well-folded, yet dynamic, protein suggesting successful sample optimization. The use of methyl-labeling is also advantageous as the lack of exchange of methyl protons with bulk water allows us to conduct these experiments at pH 8.5 without loss of signal. Collecting spectra at pH 8.5 thus improves the recovery during buffer exchange. While not tested yet, it is possible that looking at ${}^{1}$H-${}^{15}$N correlations in casp-6 FL C163S may prove an effective method for studying the disordered prodomain and flexible intersubunit linker of the casp-6 zymogen. This approach, however, would require optimizing a method for exchanging casp-6 into a lower pH buffer.

The optimization of this protocol was initiated with the sole intention of providing a stable sample for investigation of casp-6 structure and dynamics by NMR. This was expected to be a complicated feat due to the low reported yields of casp-6 as well as the inability to effectively concentrate casp-6 by traditional methods. Throughout the method development process, we made observations that emerged as solutions that will likely be useful to other scientists planning to study challenging proteins by NMR. Specifically, optimizing buffer conditions, as well as concentration and exchange methods for self-destructive proteins like proteases is critical for obtaining highly-concentrated yet stable samples. We hope that this sharing of our experiences in this optimization process may be able to help others as they set out on their own NMR sample preparation endeavors.

## Materials and methods

### Plasmid constructs

The construct used for wild-type casp-6 (casp-6 D179 CT) was designed to express a fully mature form of the enzyme (residues 24–179 in the large subunit and 194–293 in the small subunit attached to a C-terminal His$_6$ tag) in a pET11a vector and has been described previously as a casp-6 D179 CT construct [16]. The full-length construct of casp-6 (casp-6 FL C163S) is designed to express residues 1–293 with a C-terminal His$_6$ tag in a pET11a vector as described previously [19].

## Transformation

The selected DNA construct was transformed into *E. coli* BL21 DE3 electrocompetent cells via electroporation before being diluted with 2xYT broth and growth overnight on an LB agar plate containing the resistant antibiotic (100 μg/mL ampicillin for all casp-6 constructs).

## Protein overexpression in M9 and 2x M9 minimal media

Protein overexpression parameters that were not optimized in this study, including isopropyl β-D-1-thiogalactopyranoside (IPTG) concentration, growth temperature, induction time, and induction temperature were performed as previously described [16] or according to standard isotopic-labeling procedures. For isotopically-labeled protein, seed cultures were prepared by inoculating 5 mL of LB media with a single colony and allowing it to grow at 37°C until an optical density at 600 nm ($OD_{600}$) of 0.6 was reached (~3.5 hours). 1 L of M9 minimal media contained 6.78 g/L $Na_2HPO_4$, 3 g/L $KH_2PO_4$, and 0.5 g/L NaCl while 2x M9 minimal media contained 13.56 g/L $Na_2HPO_4$, 6 g/L $KH_2PO_4$, 1 g/L NaCl. Both contained 2 g/L glucose, 1 g/L $NH_4Cl$, with the appropriate antibiotic (100 μg/mL ampicillin), as well as 10 mg/L thiamine, 10 mg/L nicotinic acid, 10 mg/L calcium pantothenate, 10 mg/L biotin, 2 mM $MgSO_4$, 150 μM $CaCl_2$, 15 mM $Na_2SO_4$, and 200 μM $FeCl_3$ (S1 Table). To 1 L 2x M9 was added 250 μL of the $OD_{600}$=0.6 seed culture. The 1 L culture was shaken at 30°C until an $OD_{600}$ of 0.7 was reached (~16 hours). To express protein with isotopically-labeled isoleucines, once an $OD_{600}$ of 0.7 was reached, isotopically-labeled precursor and/or amino acid was added (60 mg/L α-ketobutyric acid sodium salt (methyl-$^{13}$C) for $^{13}$C-Ileδ1-methyl) (Cambridge Isotope Laboratories) as reported by Kerfah et al. [58] One hour after addition of the precursor and/or amino acid or once $OD_{600}$ of 1.1 was reached, casp-6 expression was induced with isopropyl β-D-1-thiogalactopyranoside (IPTG) to a final concentration of 1 mM. Overexpression was then carried out at 20°C for 18 hours before harvesting by centrifugation at 4°C and 4,696 x g for 10 minutes. The cell pellet was then frozen and stored at −80°C prior to purification.

## Protein overexpression in 2xYT nutrient-rich media

For unlabeled protein, seed cultures were prepared by inoculating a 50 mL seed culture with a single colony and allowed to grow at 37°C for 16 hours. Large cultures of 2xYT media (16 g/L tryptone, 10 g/L yeast extract, and 5 g/L NaCl) were inoculated with 1−5 mL of the dense seed culture (200−1,000-fold dilution) and shaken at 37°C. Once an $OD_{600}$ of 0.9 was achieved (typically about 4 hours), the cultures were induced with isopropyl β-D-1-thiogalactopyranoside (IPTG) at a final concentration of 1 mM. Overexpression, cell pellet harvesting, and storage was then carried out as described for minimal media cultures.

## Casp-6 Nickel-affinity and anion-exchange purification

Cell pellets were thawed and homogenized in lysis buffer (50 mM Tris pH 8.5, 300 mM NaCl, 50 mM imidazole, and 5% glycerol) prior to being lysed in a Microfluidics™ microfluidizer at 15,000 psi. The lysed cells were separated from cellular debris by centrifugation at 4°C and 18,061 x g for 45 minutes to yield a clear supernatant. Lysate was then loaded onto a lysis buffer-equilibrated 5 mL Ni-charged HisTrap™ HP column (Cytiva Life Sciences) at a flow rate of 3 mL/min. The bound protein was then washed with 10 column volumes (CV) of lysis buffer at 5 mL/min and eluted by a step gradient with 5 CV of 73.5% lysis buffer and 26.5% Ni elution buffer (50 mM Tris pH 8.5, 300 mM NaCl, 1 M imidazole, and 5% glycerol) at 1 mL/min. Protein-containing fractions were then diluted 9-fold with anion-exchange buffer A (20 mM Tris pH 8.5, 2 mM DTT, and 5% glycerol) to lower the salt concentration, before being loaded onto a buffer A-equilibrated 5 mL HiTrap™ HP Q Anion-Exchange column (Cytiva Life Sciences) at 3 mL/min. The protein was then washed with 5 CV of buffer A at 5 mL/min. Elution began with a linear gradient to 15% v/v buffer B (20 mM Tris pH 8.5, 1 M NaCl, 2 mM DTT, and 5% glycerol) at 3 mL/min over 4 CV, followed by a step gradient to 23% buffer B at 1 mL/min for 2 CV at which point

casp-6 began eluting, and finally a linear gradient to 70% buffer B at 4 mL/min over 3 CV to clean the column. Protein-containing fractions were then pooled by concentration and purity and either aliquoted and frozen at −80°C or concentrated in an Amicon® 10-kDa cutoff spin filter from Millipore Sigma at 4°C and 3000 x g.

### NMR sample buffer exchange

Methyl-labeled samples for NMR data collection were exchanged into a fully deuterated buffer containing 20 mM d-Tris pH 8.5, 200 mM NaCl, 5% $d_6$-glycerol, and 10 mM d-DTT in 100% $D_2O$. Buffer exchange was carried out using a NAP-5 or PD-10 Sephadex-G25 desalting column (Cytiva Life Sciences) according to the manufacturer's instructions with the following changes. Elution from the desalting column was collected in 5–10 drop fractions in a 96-well plate. Fractions were then analyzed by SDS-PAGE to assess purity and determine pools. Final pools were aliquoted at 170 µL and frozen at −80°C. All deuterated reagents were purchased from Cambridge Isotope Laboratories.

### Casp-6 activity assays

Casp-6 initial rates were determined via cleavage of a fluorogenic peptide substrate VEID-amc (Cayman Chemical), from which 7-aminomethylcoumarin (amc) acts as the fluorogenic reporter. Enzyme was diluted using each of the buffers in S2 Table to test the effects of various additives and NaCl concentrations. These reactions were carried out at either 100 µL or 30 µL reaction volumes in technical duplicates with an enzyme concentration of 100 nM and a substrate concentration of 60 µM. Upon addition of the diluted enzyme to the substrate, reactions were monitored for 7 min at 37°C with $\lambda_{Ex}/\lambda_{Em}$ of 365 nm/495 nm in a SpectraMax M5 spectrophotometer.

### Data and statistical analysis

All data analysis and visualization was performed using R (version 4.4.1) in RStudio. Data were cleaned and reshaped using the dplyr, tidyr, and reshape2 packages. Bar plots with error bars representing mean ± standard deviation were generated using ggplot2, and all figures were saved as high-resolution JPEGs. Comparisons of protein yield, percent recovery, and enzymatic activity were conducted using two-sample or grouped t-tests implemented via the rstatix package. Where applicable, multiple comparisons were corrected using the Benjamini-Hochberg (BH) method as this was found suitable for analyzing exploratory trends. Statistical significance was visualized on plots using stat_pvalue_manual from ggpubr, with significance thresholds of $p < 0.05$ (*), $p < 0.01$ (**), $p < 0.001$ (***), and $p < 0.0001$ (****). Percent recovery from buffer exchange was assessed across buffer identity, salt concentration, reducing agents (e.g., DTT), presence of glycerol, and column type. Enzymatic activity (VEIDase) was quantified by calculating the linear slope of initial fluorescence signal over time, and comparisons were made between different buffer compositions, salt concentrations, and additive conditions. Duplicate measurements were averaged for rate calculations, and comparisons were made both within and across buffer conditions to assess the influence of specific additives and ionic strength on activity.

### NMR data collection

Samples (S3 Table) for 1-dimensional $^1H$ NMR contained 30–70 µM protein for casp-6 D179 CT or 140 µM protein for casp-6 FL C163S. The protein concentration in NMR was dependent on two factors: the yield following purification and the minimal concentration required to obtain sufficient NMR signal. Casp-6-containing buffers consisted of either 20 mM d-Tris pH 7.5, 50 mM HEPES pH 7.5, or 100 mM phosphate pH 7.4 with 120 mM NaCl, 2–10 mM d-DTT, and 5% $d_6$-glycerol, unless otherwise stated. NMR spectra were collected on a 600 MHz Bruker Avance III spectrometer at 20°C with a helium-cooled cryoprobe at the University of Massachusetts Amherst. $^1H$ NMR spectra were collected using an excitation sculpting pulse program for water suppression [59]. Prior to each data collection, the 90° pulse length and power for water suppression were optimized for best signal intensity. Spectra were collected with an FID containing 32768 points

and 128–4096 scans, depending on protein concentration. All data were collected and processed using TopSpin 3.6. The $^1$H-$^{13}$C heteronuclear multiple quantum coherence (HMQC) spectra of Ileδ1-$^{13}$CH$_3$ casp-6 D179 CT in 20 mM d-Tris pH 8.5, 200 mM NaCl, 5% d$_6$-glycerol, 10 mM d-DTT and 100% D$_2$O (S3 Table) were collected on a 600 MHz Bruker Avance III spectrometer at 20°C with a helium-cooled cryoprobe at the University of Massachusetts Amherst for the stability tests (S2 Fig) and a 900 MHz Bruker Avance III HD spectrometer at 35°C with a helium-cooled cryoprobe at NMRFAM (Fig 7). All $^1$H-$^{13}$C HMQC spectra used a band-Selective Optimized-Flip-Angle Short-Transient (SOFAST) pulse sequence developed by Schanda et al [60]. Higher temperature was used for the SOFAST spectrum in Fig 7 to increase signal-to-noise. For each sample the optimal 90° pulse length for proton frequency was determined prior to data collection, and a 1D $^1$H spectrum was collected before and after 2D spectrum collection to ensure protein stability throughout. The $^1$H-$^{13}$C HMQC spectra were collected for 128 scans with an FID containing 1024 complex points in the $^1$H dimension and 128 complex points in the $^{13}$C dimension. The HMQC spectra were processed with TopSpin (600 MHz) or NMRPipe (900 MHz) using standard processing routines and analyzed with NMRFAM-Sparky.

## Supporting information

**S1 Fig. Effect of varying induction OD$_{600}$ on casp-6 purity.** Increasing OD$_{600}$ did not appear to increase remaining by-products in the final purified casp-6. Fractions from the purification include cell debris after cell lysis (CD), cellular lysate (Lys), Ni column flow-through (NiFT), Ni column wash, Ni column elution (Ni Elu), anion-exchange flow-through (QFT), anion-exchange wash (Q Wash), and individual fractions from anion-exchange.
(TIF)

**S2 Fig. Casp-6 remains stable at 25°C for at least five days under optimized conditions.** A series of $^1$H-$^{13}$C HMQC spectra for Ileδ1-$^{13}$CH$_3$-labeled casp-6 D179 CT at 25°C and 600 MHz show no significant loss in signal intensity for up to 120 hours, suggesting a stable and soluble sample throughout this period.
(TIF)

**S1 Table. Composition of minimal media and 2xYT nutrient-rich media.** Minimal media as outlined here, is carefully composed of specific salts, minerals, vitamins, a single carbon source, and a single nitrogen source, whereas the complex ingredients in 2xYT provide an array of nutrients.
(DOCX)

**S2 Table. Various buffer conditions tested to assess casp-6 activity.** Buffer combinations used to assess casp-6 activity by cleavage of VEID-amc fluorogenic substrate.
(DOCX)

**S3 Table. Sample descriptions and spectrum parameters for NMR experiments.** A tabular presentation of the sample conditions, nucleus/experiment, solvent, solvent suppression, reference, and temperature information for the presented NMR spectra.
(DOCX)

**S1 Protocol. Protocol of casp-6 NMR sample preparation.** A step by step procedure of protein expression, purification, and buffer exchange for isotopically-labeled casp-6 NMR samples.
(DOCX)

**S1 Data. Folder including all processing scripts, assay data, excel sheets, NMR spectrum files, and Purification Methods.** Data are categorized by its respective main text figure. Files are titled with a description of contents. Exported text instructions and images for optimized purification protocols are in the *Purification Methods* folder.
(ZIP)

**S1 Raw Images. File containing all uncropped gel images.** Original uncropped images of the caspase-6 purification and exchange gels.

(PDF)

## Acknowledgments

We acknowledge the helpful discussions and feedback we received from the UMass Structural Biology Joint Group Meeting.

## Author contributions

**Conceptualization:** Nathanael J. Kuzio, Jasna Fejzo, Jeanne A. Hardy.

**Data curation:** Nathanael J. Kuzio.

**Formal analysis:** Nathanael J. Kuzio, Marco Tonelli, Jeanne A. Hardy.

**Funding acquisition:** Jeanne A. Hardy.

**Investigation:** Nathanael J. Kuzio, Marco Tonelli.

**Methodology:** Nathanael J. Kuzio, Marco Tonelli, Jasna Fejzo, Jeanne A. Hardy.

**Resources:** Jasna Fejzo, Jeanne A. Hardy.

**Supervision:** Jasna Fejzo, Jeanne A. Hardy.

**Validation:** Nathanael J. Kuzio.

**Visualization:** Nathanael J. Kuzio, Jeanne A. Hardy.

**Writing – original draft:** Nathanael J. Kuzio.

**Writing – review & editing:** Nathanael J. Kuzio, Marco Tonelli, Jasna Fejzo, Jeanne A. Hardy.

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
