## [Decision Letter · Decision Letter 0]

15 Aug 2025

Dear Dr. Hardy,

Thank you for submitting your manuscript to PLOS ONE. After careful consideration, we feel that it has merit but does not fully meet PLOS ONE’s publication criteria as it currently stands. Therefore, we invite you to submit a revised version of the manuscript that addresses the points raised during the review process.

We look forward to receiving your revised manuscript.

Kind regards,

Ajit Prakash, PhD

Academic Editor

PLOS ONE

Journal Requirements:

2. We note that this submission includes NMR spectroscopy data. We would recommend that you include the following information in your methods section or as Supporting Information files:

1) The make/source of the NMR instrument used in your study, as well as the magnetic field strength. For each individual experiment, please also list: the nucleus being measured; the sample concentration; the solvent in which the sample is dissolved and if solvent signal suppression was used; the reference standard and the temperature.

2) A list of the chemical shifts for all compounds characterised by NMR spectroscopy, specifying, where relevant: the chemical shift (δ), the multiplicity and the coupling constants (in Hz), for the appropriate nuclei used for assignment.

3)The full integrated NMR spectrum, clearly labelled with the compound name and chemical structure.

We also strongly encourage authors to provide primary NMR data files, in particular for new compounds which have not been characterised in the existing literature. Authors should provide the acquisition data, FID files and processing parameters for each experiment, clearly labelled with the compound name and identifier, as well as a structure file for each provided dataset. See our list of recommended repositories here: https://journals.plos.org/plosone/s/recommended-repositories .

Reviewers' comments:

Reviewer's Responses to Questions

**Comments to the Author**

1. Is the manuscript technically sound, and do the data support the conclusions?

Reviewer #1: Yes

Reviewer #2: Yes

2. Has the statistical analysis been performed appropriately and rigorously?

Reviewer #1: Yes

Reviewer #2: Yes

3. Have the authors made all data underlying the findings in their manuscript fully available?

Reviewer #1: Yes

Reviewer #2: Yes

4. Is the manuscript presented in an intelligible fashion and written in standard English?

Reviewer #1: Yes

Reviewer #2: Yes

Reviewer #1: Summary:

This manuscript describes the successful optimization of an NMR sample preparation protocol for Caspase-6, a self-degrading protease that has been historically difficult to study using solution NMR. The authors systematically test different growth, purification, and buffer exchange conditions to obtain stable, concentrated, isotopically labeled samples. Their optimized protocol improves yield and spectral quality by eliminating steps that cause protein loss, such as anion exchange and spin filtration. The study also evaluates multiple constructs and conditions to find one that is NMR-compatible. This work offers practical guidance for other researchers working on similarly unstable or low-yield proteins.

Strengths:

1. The study addresses a longstanding technical challenge in the preparation of an NMR-compatible form of Caspase-6, a self-cleaving and aggregation-prone protease.

2. The experimental design is thorough and data-driven leading to a final protocol that is simple, reproducible, and avoids problematic steps like anion exchange and centrifugal concentration.

3. The authors choose an appropriate construct (D179CT) and NMR approach (Ileδ1 13CH3 methyl HMQC) for a first step in structural work.

4. The manuscript is well organized, and the conclusions are well supported by experimental outcomes.

5. The work is broadly applicable to researchers preparing dynamic or unstable proteins for biophysical characterization.

Major Comments:

1. The manuscript states that Ni-affinity purification alone provided sufficient purity and improved yield. However, this claim would be stronger if the authors included direct comparisons of yield, concentration, and purity between samples that did and did not undergo anion exchange. An SDS-PAGE image or protein quantification comparing the two workflows would be useful.

2. The authors should clarify in the Introduction or early Results that spin filters caused significant sample loss and were avoided throughout the workflow. Including specific protein recovery data comparing desalting columns to centrifugal filters will support this decision and strengthen the general takeaway.

3. The various optimizations are scattered throughout the text. A summary comparing the initial and final workflows with numerical data for yield (mg/L), concentration (µM), recovery %, and NMR suitability would help readers quickly understand the benefits of each change.

4. The authors should report how many Ileδ1 methyl peaks were observed in the HMQC spectrum relative to the expected number based on sequence. This will help assess how well-folded and NMR-ready the sample is.

5. The authors should comment on signal-to-noise ratio, chemical shift dispersion, and linewidth in the methyl HMQC spectrum. Even if qualitative, this information will support claims of spectral quality and sample integrity.

6. The authors should assess how stable the Caspase-6 samples are over a 24- to 48- hour period at room temperature or 25°C. Overlays of spectra at different time points or a description of peak loss or aggregation would help others plan experiments using this protocol.

7. If the authors have successfully expressed and collected a 1H-15N HSQC spectrum for the full-length C163S variant, it would be valuable to include this data, even if unassigned. This will strengthen the claim that the optimized protocol supports analysis of other constructs.

8. The manuscript states that these findings may apply to other proteases. The authors should be more specific about which steps, such as choice of Tris pH 8.5, inclusion of DTT, use of glycerol, or buffer exchange strategy, are likely to benefit other unstable or aggregation-prone proteins.

Minor Comments:

1. Standardize the use of “Caspase-6” or “Casp-6” throughout the manuscript for consistency.

2. Ensure that all figures (particularly Fig 2A, 2C, 3B, 4B, 6A-D) include properly labeled axes, units, and legends. Clarify what the error bars represent (SD or SEM), and indicate the number of replicates (n).

3. In the Methods section, clarify whether IPTG concentration, growth temperature, and induction timing were fixed based on prior studies or tested as part of this optimization.

4. All supplemental materials (S1 Protocol, S1 Fig, S2 Table) should be cited at appropriate points in the main text where their content is relevant. Avoid listing them only in the Supporting Information section.

5. In the abstract, revise the sentence “it remained impossible to analyze caspase-6 by NMR” to something more direct, such as “Prior attempts to analyze Caspase-6 by NMR failed due to challenges in sample preparation.”

Reviewer #2: The section is scientifically sound and well-documented, but it would benefit from:

Condensing repeated explanations,

Providing brief quantitative context when referencing figures,

Structuring into smaller paragraphs for readability,

Emphasizing the key conclusion more prominently.

**Do you want your identity to be public for this peer review?** For information about this choice, including consent withdrawal, please see our Privacy Policy

Reviewer #1: No

Reviewer #2: **Yes: ** DR SAI KUMAR BADAM

---

## [Author Response · Author response to Decision Letter 1]

24 Sep 2025

We thank the reviewers for their attention and time in providing a very comprehensive review. We have provided a thorough and point-by-point response in the "Response to Reviewers" document.

---

## [Decision Letter · Decision Letter 1]

7 Nov 2025

An NMR Sample Preparation Case Study: Considerations for the Self-Destructive Protease Caspase-6

PONE-D-25-34737R1

Dear Dr. Hardy,

We’re pleased to inform you that your manuscript has been judged scientifically suitable for publication and will be formally accepted for publication once it meets all outstanding technical requirements.

Kind regards,

Ajit Prakash, PhD

Academic Editor

PLOS ONE

Additional Editor Comments (optional):

Reviewers' comments:

Reviewer's Responses to Questions

**Comments to the Author**

Reviewer #1: All comments have been addressed

Reviewer #2: All comments have been addressed

2. Is the manuscript technically sound, and do the data support the conclusions?

Reviewer #1: Yes

Reviewer #2: Yes

3. Has the statistical analysis been performed appropriately and rigorously?

Reviewer #1: Yes

Reviewer #2: Yes

4. Have the authors made all data underlying the findings in their manuscript fully available?

Reviewer #1: Yes

Reviewer #2: Yes

5. Is the manuscript presented in an intelligible fashion and written in standard English?

Reviewer #1: Yes

Reviewer #2: Yes

Reviewer #1: (No Response)

Reviewer #2: (No Response)

**Do you want your identity to be public for this peer review?** For information about this choice, including consent withdrawal, please see our Privacy Policy

Reviewer #1: No

Reviewer #2: **Yes: ** Dr. SAI KUMAR BADAM

---

## [Editor Report · Acceptance letter]

PONE-D-25-34737R1

PLOS ONE

Dear Dr. Hardy,

I'm pleased to inform you that your manuscript has been deemed suitable for publication in PLOS ONE. Congratulations! Your manuscript is now being handed over to our production team.

Kind regards,

on behalf of

Dr. Ajit Prakash

Academic Editor

PLOS ONE